# Molecular Mimicry between SARS-CoV-2 Proteins and Human Self-Antigens Related with Autoimmune Central Nervous System (CNS) Disorders

**DOI:** 10.3390/microorganisms11122902

**Published:** 2023-12-01

**Authors:** Elisa Gouvea Gutman, Renan Amphilophio Fernandes, Jéssica Vasques Raposo-Vedovi, Andreza Lemos Salvio, Larissa Araujo Duarte, Caio Faria Tardim, Vinicius Gabriel Coutinho Costa, Valéria Coelho Santa Rita Pereira, Paulo Roberto Valle Bahia, Marcos Martins da Silva, Fabrícia Lima Fontes-Dantas, Soniza Vieira Alves-Leon

**Affiliations:** 1Translational Neuroscience Laboratory (LabNet), Biomedical Institute, Federal University of the State of Rio de Janeiro, Rio de Janeiro 20211-030, RJ, Brazil; gutman811@gmail.com (E.G.G.); renanyfernandes@gmail.com (R.A.F.); jessicavasquesr@gmail.com (J.V.R.-V.); andrezaslemos@gmail.com (A.L.S.); larissa.lariad@gmail.com (L.A.D.); 2Clinical Medicine Post-Graduation Program, Federal University of Rio de Janeiro, Rio de Janeiro 21941-913, RJ, Brazil; 3Department of Neurology, Clementino Fraga Filho University Hospital, Federal University of Rio de Janeiro, Rio de Janeiro 21941-913, RJ, Brazil; caiofariatardin@gmail.com (C.F.T.); valeriacsrp@yahoo.com (V.C.S.R.P.); marcosm@hucff.ufrj.br (M.M.d.S.); 4Morphological Sciences Post-Graduation Program, Federal University of Rio de Janeiro, Rio de Janeiro 21941-913, RJ, Brazil; vincius96@gmail.com; 5Department of Radiology, Clementino Fraga Filho University Hospital, Federal University of Rio de Janeiro, Rio de Janeiro 21941-913, RJ, Brazil; vallebahia@gmail.com; 6Department of Pharmacology, Institute of Biology, Rio de Janeiro State University, Rio de Janeiro 20950-000, RJ, Brazil

**Keywords:** COVID-19, SARS-CoV-2, molecular mimicry, immune tolerance, autoimmune disorders, central nervous system

## Abstract

SARS-CoV-2 can trigger autoimmune central nervous system (CNS) diseases in genetically susceptible individuals, a mechanism poorly understood. Molecular mimicry (MM) has been identified in other viral diseases as potential triggers of autoimmune CNS events. This study investigated if MM is the process through which SARS-CoV-2 induces the breakdown of immune tolerance. The frequency of autoimmune CNS disorders was evaluated in a prospective cohort with patients admitted to the COVID-19 Intense Care Unity (ICU) in Rio de Janeiro. Then, an in silico analysis was performed to identify the conserved regions that share a high identity between SARS-CoV-2 antigens and human proteins. The sequences with significant identity and antigenic properties were then assessed for their binding capacity to HLA subtypes. Of the 112 patients included, 3 were classified as having an autoimmune disorder. A total of eleven combinations had significant linear and three-dimensional overlap. NMDAR1, MOG, and MPO were the self-antigens with more significant combinations, followed by GAD65. All sequences presented at least one epitope with strong or intermediate binding capacity to the HLA subtypes selected. This study underscores the possibility that CNS autoimmune attacks observed in COVID-19 patients, including those in our population, could be driven by MM in genetically predisposed individuals.

## 1. Introduction

The recent emergence of the severe acute respiratory syndrome coronavirus 2 (SARS-CoV-2) virus, responsible for the disease caused by the coronavirus disease 2019 (COVID-19), has already affected nearly seven hundred million individuals worldwide, with 6,919,573 deaths until September 2023 [1]. As evidenced in recent years, the occurrence of viral epidemics characterized by unpredictable clinical outcomes has been highly frequent on a global scale [2]. Therefore, the end of the public health emergency of international concern declared on 5 May 2023 by the World Health Organization (WHO) does not mean that COVID-19 has ceased to be a serious public health problem. Extensive studies have provided strong evidence of the frequent occurrence of neurological manifestations that resemble clinical patterns seen in autoimmune para- or post-infectious diseases triggered by SARS-CoV-2 [3,4]. This pattern is seen in other viral infections, such as Zika and Chikungunya infection [5,6,7]. Moreover, the hypothesis that viral infections can act as triggers for the development of autoimmune diseases is not a new one [8]. Due to the association between infection with other coronaviruses and autoimmunity, it is reasonable to assume that there is a connection between SARS-CoV-2 infection and certain autoimmune diseases that will be diagnosed later [9,10,11].

It is already well discussed in the literature that viral diseases have been identified as potential triggers of inflammatory demyelinating diseases (IDDs) and autoimmune encephalitis (AE) [12,13]. Numerous demyelinating disorders such as multiple sclerosis (MS), neuromyelitis optica spectrum diseases (NMOSDs), acute disseminated encephalomyelitis (ADEM), myelitis, and myelin oligodendrocyte glycoprotein antibody-associated disease (MOGAD) have been described as post and para-infectious complications of COVID-19 [6], with the most common forms of AE following SARS-CoV-2 infection being limbic encephalitis and Anti-N-methyl-D-aspartate receptor (anti-NMDAR) encephalitis [14]. Important studies have identified the presence of autoantibodies anti-glutamic acid decarboxylase 65-kilodalton isoform (anti-GAD65) [15], anti-myelin oligodendrocyte glycoprotein (anti-MOG), and others [16,17,18,19].

Several hypotheses have been proposed to explain the molecular basis of the loss of immune tolerance and induction of autoimmune mechanisms, including hyperinflammation syndrome caused by SARS-CoV-2, molecular mimicry (MM) by viral proteins, immune cell activation through bystander effect, the release of autoantigens from virus-damaged tissues, lymphocyte activation mediated by superantigens, and epitope spreading [20,21,22,23]. Some recent studies on MM have shown similarities between SARS-CoV-2 protein sequences and human proteins found in multiple organs/tissues (neurological, vascular, and cardiac), indicating the potential for cross-reactive immune recognition of these regions by T cells and antibodies produced by B cells [20,21,24,25]. However, the true spectrum of autoimmune conditions, their pathophysiology, prevalence, and the risk of their development in individuals after SARS-CoV-2 infection remain unknown, representing just the tip of the iceberg.

In this study, we intend to investigate the potential role of MM between SARS-CoV-2 antigens and human autoantigens of CNS autoimmune diseases.

## 2. Materials and Methods

### 2.1. Study Population

A prospective cohort study was performed with patients admitted to the COVID-19 Intense Care Unity (ICU) of Clementino Fraga Filho University Hospital in Rio de Janeiro, RJ, Brazil. This work was approved by the National Council for Ethics in Research (CAAE 33659620.1.1001.5258) and accompanied afterward on the post-COVID ambulatory of the same hospital. All subjects signed an informed consent agreeing to participate in this research. From 2020 to 2022, COVID-19 patients were evaluated with complete physical and neurological examinations and searched for CNS autoimmune diseases. They were also actively asked for long COVID symptoms, with a focus on cognitive impairment. In addition, all patients were assessed for objective alterations with the following neurocognitive battery tests: the Symbol Digit Modalities Test (SDMT) and Montreal Cognitive Assessment (MoCA). Patients with previously known CNS autoimmune diseases or neurocognitive disturbances were excluded from the study.

### 2.2. Linear Sequence Analysis

Peptide sharing between SARS-CoV-2 antigens and autoantigens was analyzed in accordance with França et al. (2023) [26]. Briefly, a viral polyprotein library was constructed using the major viral antigens reported in the literature and protein sequences available in the National Center for Biotechnology Information (NCBI) Protein Reference Sequences https://www.ncbi.nlm.nih.gov/protein (accessed on 19 January 2023) (Table 1). An extended research study was conducted to build an autoantigen library from the UniProtKB Database Release 2023_01 www.uniprot.org/ (accessed on 19 January 2023) based on results from PubMed [20,24,25] of autoantigen related to demyelinating brain diseases and autoimmune encephalitis (Table 2).

The sequence alignment was performed using EMBOSS WATER https://www.ebi.ac.uk/Tools/psa/emboss_water/ (accessed on 21 January 2023), an online server that uses the Smith–Waterman algorithm (modified for speed enhancements) to calculate the local alignment of two sequences, narrowing down the regions with more identity [27]. The chosen regions were analyzed for linear homology between the identified human proteins to SARS-CoV-2 proteins using BLAST+ 2.13.0 - BLASTp https://blast.ncbi.nlm.nih.gov/Blast.cgi (accessed on 21 January 2023) [28]. We used the default BLASTp algorithm parameters to consider a significant result [29].

### 2.3. Three-Dimensional Comparative Modelling

The combinations that shared significant linear identity according to BLASTp were then investigated for three-dimensional similarities. The three-dimensional models were built using the Swiss Model, an online modeling server https://swissmodel.expasy.org/ (accessed on 23 January 2023). The template modeling scores (TM-scores) and root mean square deviation (RMSD) of the SARS-CoV-2 antigens and autoantigens three-dimensional overlap were calculated using TM-Align Version 20140601 https://seq2fun.dcmb.med.umich.edu//TM-align/ (accessed on 21 January 2023), an algorithm for sequence-independent protein structure comparisons. TM-align first generates optimized residue-to-residue alignment based on structural identity using heuristic dynamic programming iterations. The TM-score value scales the structural identity varying from 0.0 to 1.0, where scores below 0.3 correspond to randomly chosen unrelated proteins, while those higher than 0.5 assume generally the same fold between two structures, based on the Protein Data Bank (PDB) [30]. The RMSD considers the root-mean-square distance between corresponding residues and is calculated after an optimal rotation of one structure to another.

### 2.4. Antigenic Prediction

To confirm whether the SARS-CoV-2 sequences studied have antigenic properties, VaxiJen version 2.0 http://www.ddgpharmfac.net/vaxijen/VaxiJen/VaxiJen.html (accessed on 3 August 2023), was used. A threshold antigenic score of 0.4 was defined to filter probable non-antigenic sequences. The Vaxijen server performs alignment-independent prediction, which is based on auto cross-covariance transformation of protein sequences into uniform vectors of principal amino acid properties.

### 2.5. Search for Potential T Cell Epitopes

The sequences with significant TM-Score and antigenic properties were used as the input in a neural network–based algorithm to predict T cell epitopes showing binding capacity to human leukocyte antigen (HLA) subtypes using the Immune Epitope Database and Analysis (IEDB) Major Histocompatibility Complex-I (MHC-I) Binding Predictions http://tools.iedb.org/mhci/ (accessed on 6 September 2023), and Major Histocompatibility Complex-II (MHC-II) Binding Predictions http://tools.iedb.org/mhcii/ (accessed on 6 September 2023) resource. This approach enabled the distinction of T cell epitopes recognized by HLA. As HLAs exhibit high polymorphism, we chose HLA variants from MHC-I and MHC-II, with known associations with CNS autoimmune diseases. The representatives used were HLA-I A*31:01, HLA-I B*07:02, HLA-II DRB1*1501, HLA-II DQA1*0102-HLA-II DQB1*0602, and HLA-II DRB1∗03:01.

The prediction method used was the stabilization matrix alignment method (SMM-align) version 1.1. The predicted output is given in units of half maximal inhibitory concentration (IC_50_nM). Therefore, a lower number indicates higher affinity. As a rough guideline, peptides with IC_50_ values <50 nM are considered high affinity, <500 nM intermediate affinity, and <5000 nM low affinity. Since most known epitopes have high or intermediate affinity, we only considered a noteworthy result for the epitopes with <500 nM [31].

## 3. Results

### 3.1. Study Population

A total of 112 patients were evaluated. The mean age was 65.95 (17–95), and 58 (51.78%) were women. Among them, three were classified as having IDD. As our population consisted mainly of patients from the first and second wave of the COVID-19 pandemic, and all of them were hospitalized in the ICU, 51 (45.53%) patients progressed to death during or nearly after the hospitalization. The patients with IDD survived and remained with controlled disease for one year after the episodes. Among the survivors, 32 patients (52.45%) presented cognitive complaints during long COVID. Regarding the neurocognitive battery, 15 (24.59%) patients had cognitive impairment according to the 2 test results, while 20 had alteration only in SDMT and 33 had alteration only in MoCA.

#### 3.1.1. Patient 1

Patient 1, male, 24 years old, with a history of IDD in the family (mother diagnosed with MS) was admitted to the hospital presenting paresthesia in the upper and lower limbs on the left side. The Magnetic Resonance Imaging (MRI) showed two white-matter lesions hyperintense in T2 and fluid-attenuated inversion recovery (FLAIR) with gadolinium enhancement, one on the periventricular region and one on the medullary bulb transition (Figure 1). The cerebrospinal fluid (CSF) exam was negative for infections, including SARS-CoV-2, and showed oligoclonal bands with normal cell and protein count. Serum research was positive for anti-MOG antibodies through flow cytometry and negative for anti-AQP4 antibodies. Screening for metabolic and other autoimmune diseases was negative. Although asymptomatic, as part of the hospital protocol, he was tested for COVID-19 with a Polymerase Chain Reaction (PCR) test, which was positive. He was treated with intravenous glucocorticoids, evolving with complete recovery. On the long COVID assessment, he presented new symptoms and new lesions on the MRI, being diagnosed with MOGAD.

#### 3.1.2. Patient 2

Patient 2, female, 19 years old, evolved 20 days after SARS-CoV-2 m-RNA vaccination with subacute paresthesia in hands and feet, followed by urinary incontinence, visual disturbance, mental confusion, appendicular ataxia, progressive tetraparesis, and coma within days. Her Expanded Disability Status Scale (EDSS) during the acute state was 9.5. The MRI showed countless white-matter lesions hyperintense in T2/FLAIR, several with gadolinium enhancement (Figure 2). CSF exam was positive for oligoclonal bands, and negative for infections, including SARS-CoV-2. Screening for autoimmune diseases, including MOG and AQP4 antibodies, and metabolic diseases were negative. First, she received a diagnosis of ADEM following COVID-19 vaccination and was treated with pulse therapy with glucocorticoids, partially recovering from the attack. Nonetheless, after 6 months she once again evolved with a new aggressive demyelinating event, being diagnosed with MS and treated with natalizumab. The patient stabilized with the treatment, and, after a year, she recovered nearly a hundred percent (EDSS 3.0).

#### 3.1.3. Patient 3

Patient 3, female, 39 years old, with a history of psoriasis, spontaneous miscarriages, and reducing gastroplasty was admitted due to paresthesia in the right hemiface and left lower limb, as well as deviation of the labial commissure, right auricular fullness, vertigo, and headache. A diagnostic investigation with complementary tests was initiated. Eighteen days before, she had COVID-19 confirmed with PCR, with headache, cough, and sore throat. The MRI showed hyperintense T2/FLAIR oval white-matter lesions on the right cortex, periventricular region, and brainstem. (Figure 3). The CSF exam was positive for oligoclonal bands, and negative for infections, including SARS-CoV-2 and autoantibodies. Screening for autoimmune diseases, including MOG and AQP4 antibodies, and metabolic diseases were negative. At first, she was diagnosed with ADEM following SARS-CoV-2 infection. Nonetheless, three months later, she evolved with new focal symptoms and new lesions on the MRI, with gadolinium enhancement. Therefore, she was diagnosed with MS and treated with dimethyl fumarate. The disease stabilized and she stayed asymptomatic since then.

### 3.2. Sequence Identification

The extended literature research led to the selection of eight viral proteins and ten self-proteins associated with CNS demyelinating diseases and autoimmune encephalitis, listed as follows:

SARS-CoV-2 proteins: spike protein (S), envelope protein (E), leader protein/non-structural protein 1 (Nsp1), non-structural protein 2 (Nsp2), non-structural protein 3 (Nsp3), non-structural protein 13/helicase (Nsp13), ORF7a, and nucleocapsid (N) (Table 1).

Self-proteins: glutamic acid decarboxylase 65-kilodalton isoform (GAD65), myelin proteolipid protein (PLP), myelin basic protein (MBP), myelin-oligodendrocyte glycoprotein (MOG), myelin-associated glycoprotein (MAG), myelin-associated oligodendrocyte basic protein (MOBP), transaldolase, 2’,3’-Cyclic-nucleotide 3’-phosphodiesterase (CNP), aquaporin-4 (AQP4), N-methyl-D-aspartate receptor 1 (NMDAR1), and myeloperoxidase (MPO) (Table 2).

The FASTA archive of all proteins can be found in the Appendix A.

### 3.3. Linear and Three-Dimensional Analysis

A total of 80 possible arrangements were made through the bioinformatics approach to identify the sequences that shared linear and three-dimensional identity with human autoantigens of demyelinating brain diseases and autoimmune encephalitis.

The resulting arrangements were ranked based on the highest TM-scores, excluding randomly arranged and unrelated proteins (TM-score < 0.3), leaving 29 arrangements (Table 3). It is noteworthy that we only considered the SARS-CoV-2 sequences with reported linear identity on BLASTp meaningful, varying from 62.50% to 100% of identity, with significant E-score and antigenic properties according to VaxiJen (threshold antigenic score of 0.4).

Among these arrangements, eleven three-dimensional models had a significant linear and three-dimensional overlap of autoimmune CNS proteins and SARS-CoV-2 proteins (TM-score ≥ 0.5). The most similar structures were M and NMDAR1 (TM-score: 0.89), M and MPO (TM-Score = 0.73), nsp2 and NMDAR1 (TM-score = 0.69), S and MOG (TM-score = 0.63), ORF7a and MOG (TM-score = 0.62), N and MPO (TM-score = 0.59), nsp13 and GAD65 (TM-score = 0.52), nsp1 and GAD65 (TM-score = 0.52), nsp1 and MOG (TM-score = 0.50), nsp3 and MPO (TM-score = 0.50), and S and NMDAR1 (TM-score = 0.50) (Figure 1).

NMDAR1, MOG, and MPO were the self-antigens with more significant identity with SARS-CoV-2 antigens, each one with three different proteins. GAD65 also had significant identity with two virus antigens. PLP, MBP, MOBP, MAG, AQP4, and transaldolase demonstrated significant linear identity with at least one virus protein, along with three-dimensional overlap not considered random. Nonetheless, their TM-scores were below 0.5, meaning they are not on the same fold. This means that their MM is possible, yet less achievable in practice.

### 3.4. Search for Potential T Cell Epitopes

The sequences of the eleven combinations were used as the input in a neural network–based algorithm to predict their binding capacity to HLA subtypes related to autoimmune CNS diseases. All sequences presented at least one epitope with strong or intermediate binding capacity to the chosen HLA subtypes (Table 4). The binding capacity of all 29 arrangements with TM-Score > 0.3 can be seen on the Appendix A of this paper. The arrangements with the strongest binding capacities were seen with the nsp1 and GAD65 epitopes of HLA-A*31:01 (Ic50 31.44 and 27.31, respectively) and with the nsp13 and GAD65 epitopes of HLA-A*31:01 (Ic50 18.72 and 35.44, respectively).

NMDAR1 combinations had the highest number of epitopes with strong or intermediate binding capacity with two combinations binding to all four selected HLA subtypes (nsp2 and NMDAR1, and S and NMDAR1), and the same epitope had identity with different virus antigens (M and nsp2).

Interestingly, different combinations shared the same viral epitope, such as M with NMDAR1 and MPO, and nsp1 with GAD65 and MOG (Table 4). This reinforces the potential of such a region to possibly trigger an unwanted autoimmune response by mimicking distinct self-proteins.

## 4. Discussion

SARS-CoV-2 is widely studied for the generation of multi-system autoimmune reactions [32]. In this sense, the triggering of CNS autoimmune diseases seems to be a consequence of an imprecise adaptive immune system response to the presence of viral antigens. It is well known that some viruses demonstrated neurotropic features [33,34,35,36] and replication within the brain tissue, as shown by our team with the Zika virus (ZIKV) [33]. However, even ZIKV nervous system manifestation is not always associated with acute infection, and MM seems to justify these events [37,38]. Similarly, in COVID-19, viral load or severe acute infection does not seem to be the only mechanism to justify CNS involvement [39,40]. The occurrence of IDD phenotypes and encephalitis as para or postinfectious events seems to be an immune-mediated response induced by SARS-CoV-2 [41].

A large study from various global health organizations found that the incidence of autoimmune diseases was significantly higher in the COVID-19 cohort compared to the non-COVID-19 group after a 6-month follow-up period [42]. Another similar study identified a 43% higher likelihood of developing an autoimmune disease between 3 to 15 months after infection compared to a non-COVID-19 cohort [43]. Despite the progress made, cases of CNS autoimmunity after COVID-19 are rare and mainly consist of isolated case reports or case series, which provide limited information regarding clinical outcomes [44].

Although the target of such supposed autoimmune mechanisms, precisely regarding the CNS manifestations, is still not fully understood, our findings suggest that cross-reaction with selected CNS proteins associated with autoimmune brain diseases is possible to occur secondary to the immune response to SARS-CoV-2 infection. However, the risk of developing these diseases or experiencing relapses in the setting of COVID-19 remains relatively low [45]. In our cohort, three patients developed IDD following SARS-CoV-2 infection or vaccination. Since the CNS autoimmune manifestations after COVID-19 are rare, it is expected that genetic predisposition plays an essential role in the disease mechanism [46]. Despite the low frequency, the identification of IDD in such circumstances is primordial, taking into consideration the high prevalence of SARS-CoV-2 infection and the possible critical state in which the patients may encounter it. For example, our first patient evolved with EDSS 9.5, a near-death experience in a young individual with no previous comorbidity. Moreover, is important to consider SARS-CoV-2 infection as a possible trigger of IDDs because some patients present demyelinating events as the only symptom of COVID-19, as happened with our second patient.

Molecular mimicry has been described as an essential immune mechanism involved in autoimmune reactions, especially from viruses [8]. The sharing of a linear amino acid sequence or a three-dimensional conformation fit between an antigen of the virus and a host self-protein can trigger a cross-reaction from the adaptive immune system and, therefore, have a major role in initiating an autoimmune response in genetically susceptible individuals [47]. Several researchers have recognized molecular mimicry as a component of COVID-19 pathophysiology [48,49,50]. For example, Lucchese et al. observed that molecular mimicry between SARS-CoV-2 antigens and respiratory pacemaker neurons may contribute to understanding respiratory failure [51]. Hence, MM may be a key component of the immune system dysregulated response in the CNS.

In this study, 80 possible arrangements of identity among SARS-CoV-2 antigens and self-antigens related to autoimmune CNS diseases were made. Among these arrangements, eleven models had a significant linear and three-dimensional overlap of autoimmune CNS proteins and SARS-CoV-2 proteins (TM-score ≥ 0.5). NMDAR1, MOG, and MPO were the self-antigens with more significant identity with SARS-CoV-2 antigens, followed by GAD65. Notably, dysregulated serum levels of autoantibodies NMDAR, GAD65, and MOG were detected in patients with severe COVID-19 compared with healthy controls and mild COVID-19 patients [52].

MS is a classic example of an autoimmune CNS disease characterized by chronic inflammation and demyelination [53]. SARS-CoV-2 most likely acts as a precipitating factor rather than being a direct cause of MS, triggering autoimmunity in genetically predisposed individuals. In our cohort, two patients had SARS-CoV-2-related events (vaccination and infection) as the trigger for MS. Both patients were women; however, their ages, comorbidities, symptoms, and MRI lesions were considerably different, highlighting the importance of genetic predisposal and other environmental factors on the course of the disease [54].

MBP, MOBP, PLP, and MAG are myelin proteins known to be critical autoantigens in causing demyelination in CNS leading to MS [55,56]. In our study, the TM-scores among these proteins and SARS-CoV-2 antigens were low, but not considered randomly arranged, unrelated proteins. This can mean that MM among these MS autoantigens is feasible; however, the evidence is not strong. Nevertheless, MPO, a pro-oxidative enzyme associated with immune-inflammatory, oxidative stress pathways, and cortical demyelination [57], has been gaining acceptance as an important modulator of MS activity [58]. Higher-expressing MPO genotype is overrepresented in early-onset MS in women [44], and immunohistochemical analysis shows that MPO is present in microglia in and around MS lesions [44]. This study found a significant overlap of MPO among three different SARS-CoV-2 antigens (M, N, nsp3). SARS-CoV-2’s ability to mimic MPO seems to provide a greater threat for triggering new-onset MS or worsening of MS symptoms in genetically predisposed patients, as seen during the COVID-19 pandemic [59,60].

MOGAD is an emerging subset of CNS demyelinating disease [61,62], and has also been related to COVID-19 [6]. In our study, one patient evolved with MOGAD during an asymptomatic SARS-CoV-2 infection. In addition, our in-silico analysis showed that MOG shared significant linear and three-dimensional identity with three different virus antigens (S, ORF7a, and nsp1), having the most prominent overlap with the S protein. This finding agrees with recent literature, which identified anti-MOG antibodies in the acute and post-infectious phase of SARS-CoV-2 infection and COVID-19 vaccination [63,64]. Moreover, the literature has shown that diseases associated with anti-MOG almost tripled during the COVID-19 pandemic [65].

Our study highlights that NMDAR1 has three domains with significant linear and three-dimensional identity with SARS-CoV-2 antigens, including the spike protein, and all of them have the binding capacity to T-cells to be considered epitopes. It has been proposed that SARS-CoV-2’s molecular mimicry may induce anti-NMDAR encephalitis after COVID-19 [66]. This may be a key mechanism beneath CNS manifestations of COVID-19 disease and vaccination associated with anti-NMDAR antibodies [14,67].

Moreover, GAD65 had the strongest binding capacity to HLA in this work, with two different combinations of mimicry. Cases of autoimmune encephalitis associated with GAD65 have been described following SARS-CoV-2 infection [15,68]. This finding reinforces the association of SARS-CoV-2 MM and the clinical findings related to anti-GAD65 antibody.

A growing body of evidence has demonstrated the relationship between ADEM and SARS-CoV-2 infection [69]. ADEM following SARS-CoV-2 infection and vaccination have been associated with MOG [70,71,72,73] and NMDAR [74] antibodies. Interestingly, in our study, both MOG and NMDAR are associated with S protein, the most common antigen presented in SARS-CoV-2 vaccination, which may indicate the relevance of MM in post-COVID ADEM manifestations.

In this cohort, 52.45% of the COVID-19 ICU patients presented cognitive impairment during the post-acute phase of COVID-19. Notably, NMDAR, GAD65, and MPO may be involved not only in acute encephalitis or demyelinating events but also in neurocognitive and psychiatric manifestations, frequently seen in long COVID patients [66,75,76,77]. Pathological results in cognitive screening were associated with the presence of antibodies against NMDAR and GAD65 in CSF of long COVID patients [78]. The MM between these autoantibodies and SARS-CoV-2 antigens may be a prominent asset to understanding the pathogenesis of long COVID cognitive and psychiatric symptoms.

The putative epitopes of SARS-CoV-2 that form MM are rich in alanine (Ala) and leucine (Leu) (Table 4), considerably responsible for the loop conformation seen in Figure 4 [79]. Notably, an increase in Ala has been associated with several neurological diseases such as MS, Guillain–Barré syndrome, and motor neuron disease [80]. The peptide that corresponds to the glycine/alanine repeat sequence of Epstein–Barr virus nuclear antigen-1, named P62, has been found to generate cross-reactive autoantibodies in MS patients [81] and other autoimmune diseases [82,83]. Similarly, the presence of Leu on the V3 loop region of the envelope gene of HIV-infected patients was associated with dementia [84]. Also, Leu mutations in the dengue type 4 virus envelope sequence were determinants of neurovirulence in mice [85].

Genetic susceptibility seems to explain the heterogeneity of response to immune tolerance breakdown and molecular mimicry between autoantigens and viral proteins [47]. Due to the limited knowledge about genetic susceptibility to explain mechanisms involved in the pathophysiology of AE, we chose HLA variants with known associations with CNS autoimmune diseases [56,78,79,80,81,82]. Interestingly enough, all eleven combinations with significant linear and three-dimensional identity presented at least one epitope with strong or intermediate binding capacity to the chosen HLA subtypes. In this manner, the investigation of the connection between *HLA* alleles related to CNS autoimmune diseases and the MM found in this paper can strengthen the results and possibly help elucidate the pathophysiology of these manifestations.

It is worth highlighting the results regarding the S protein. The spike, or its fragments, has the ability to cross the blood-brain barrier (BBB), irrespective of the presence of the viral RNA [83]. Furthermore, some cases have reported an association between CNS demyelination events and the use of vaccines with the S protein as the main antigen for the generation of immunological memory, which has become a major concern for health authorities worldwide [84]. Thus, the MM regarding S may be more common than the others described in this article. Indeed, both MOG and NMDAR1, which presented significant linear and three-dimensional overlap with spike, have been associated with COVID-19 in a more expressive way than the other autoimmune affection [14,65,66] and have been related to COVID-19 vaccination [67,85,86,87]. One of our patients triggered IDD following COVID-19 vaccination. Although is not possible to affirm causality, MM must be considered as a possible mechanism for this phenomenon.

As a limitation of this study, it is important to mention that it is a theoretical work; however, it is based on our cohort findings and recent literature studies regarding SARS-CoV-2. In addition, it uses validated software to give results as close as possible to reality. Additionally, the study used a limited number of *HLA* alleles in the prediction of T-cell binding capacity (only the most common *HLA* alleles in the literature associated with CNS autoimmune diseases) in order to increase the specificity of the results. Thereby, it is possible that some epitopes of rarer *HLAs* were not included in this study. Further studies are needed to validate the in-silico work described here, as well as to understand the probable genetic susceptibility some individuals have that causes the development of such manifestations.

## 5. Conclusions

The presented study proposes a demonstration of possible molecular mimicry between SARS-CoV-2 antigens and CNS autoimmune self-antigens, especially MOG, NMDAR1, GAD65, and MPO, in genetically susceptible individuals. The results agree with our cohort, with three cases of IDD, and with the most recent literature. Therefore, advancing our understanding of the key mechanisms of SARS-CoV-2-mediated autoimmunity is urgent.

## Figures and Tables

**Figure 1 microorganisms-11-02902-f001:**
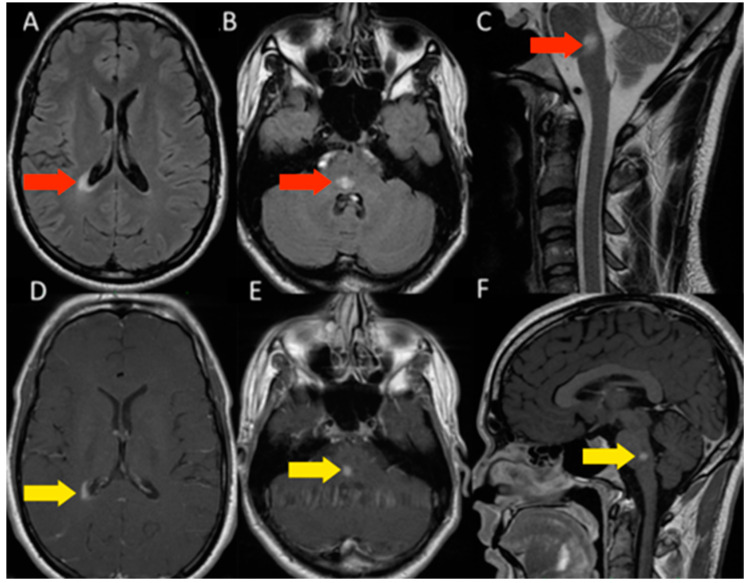
MRI of patient 1. (**A**)—axial flair sequence reveals hyperintense lesion near the posterior horn of the right lateral ventricle (red arrows). (**B**,**C**)—axial flair and sagittal T2 sequences demonstrate hyperintense lesion in the pons (red arrows). (**D**–**F**)—axial and sagittal T1 sequences with contrast demonstrating impregnation of the lesions (yellow arrows).

**Figure 2 microorganisms-11-02902-f002:**
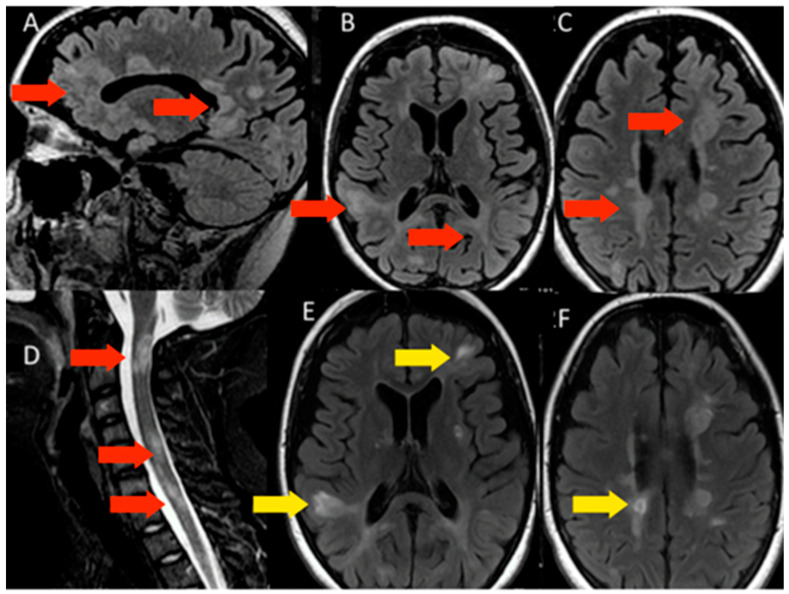
MRI of patient 2. (**A**–**D**)—sagittal and axial flair sequences and sagittal T2, showing multiple lesions (red arrows) with hypersignal, distributed in the periventricular and subcortical regions and in the spinal cord. Some of these lesions are impregnated with contrast (yellow arrows) (**E**,**F**).

**Figure 3 microorganisms-11-02902-f003:**
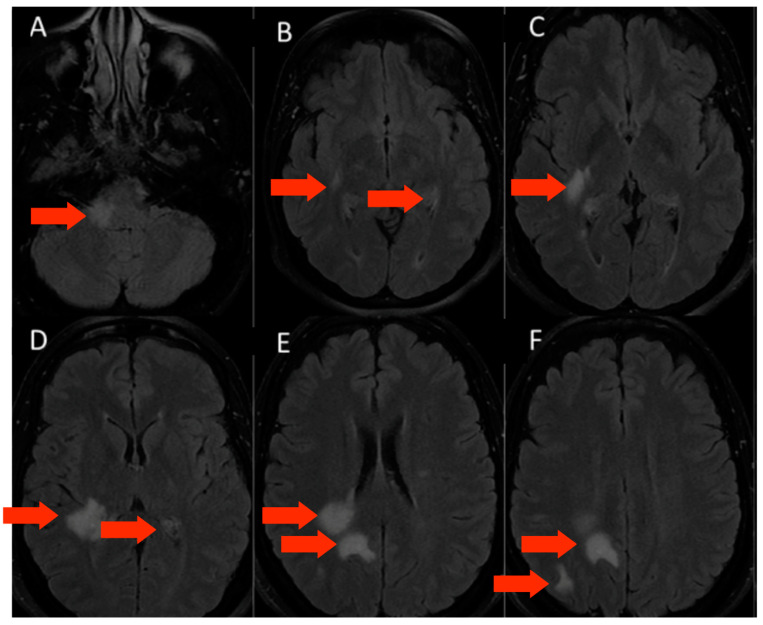
MRI of patient 3—(**A**–**F**)—flair sequence in the axial plane showing multiple lesions in the inferior cerebellar peduncle and the subcortical and periventricular regions, predominantly in the right hemisphere (red arrows).

**Figure 4 microorganisms-11-02902-f004:**
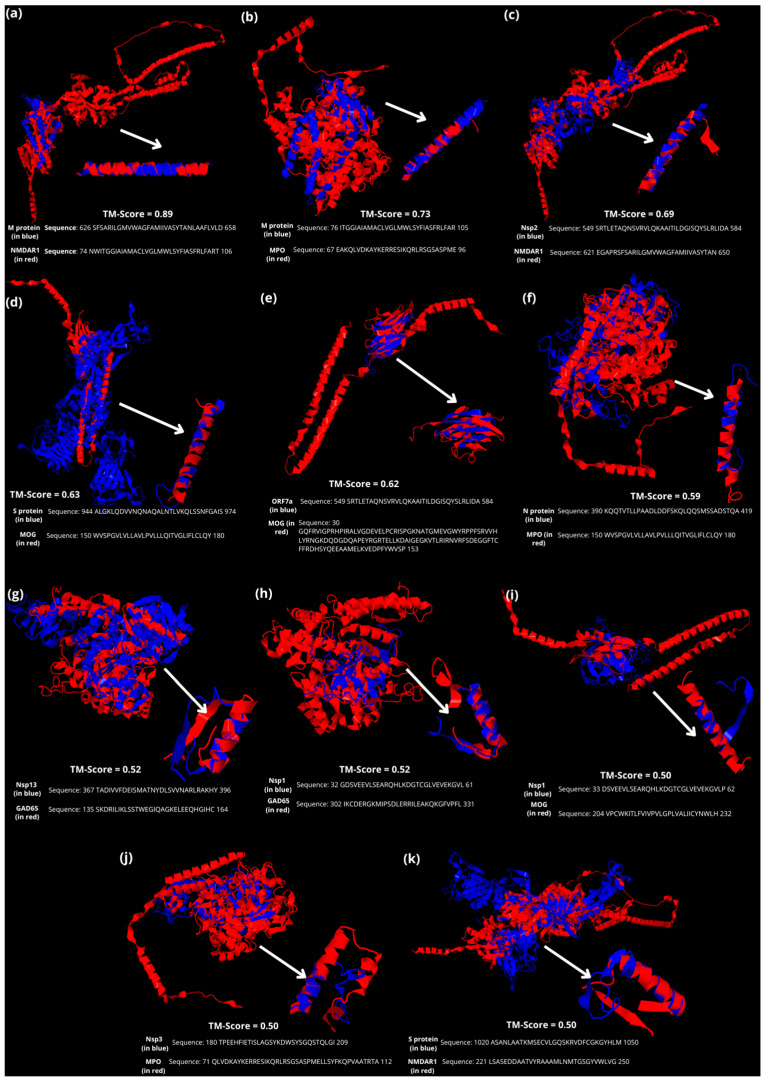
Models had a significant linear and three-dimensional overlap of autoimmune CNS proteins (in red) and SARS-CoV-2 (in blue) proteins according to TM-align, with their respective amino acid sequences. The arrows indicate the sequence region with three-dimensional overlap. (**a**) M and NMDAR1 (TM-score: 0.89). (**b**) M and MPO (TM-Score = 0.73). (**c**) nsp2 and NMDAR1 (TM-score = 0.69). (**d**) S and MOG (TM-score = 0.63). (**e**) ORF7a and MOG (TM-score = 0.62). (**f**) N and MPO (TM-score = 0.59). (**g**) nsp13 and GAD65 (TM-score = 0.52). (**h**) nsp1 and GAD65 (TM-score = 0.52). (**i**) nsp1 and MOG (TM-score = 0.50). (**j**) nsp3 and MPO (TM-score = 0.50). (**k**) S and NMDAR1 (TM-score = 0.50).

**Table 1 microorganisms-11-02902-t001:** SARS-CoV-2 proteins selected.

Protein	Number of Aminoacids	Gene	NCBI Reference Sequence	Uniprot ID
E	75aa	E	YP_009724392	P0DTC4
Nsp1	180aa	ORF1a	YP_009742608.1	P0DTD1
M	222aa	M	YP_009724393.1	P0DTC5
Nsp2	638aa	ORF1a	YP_009742609.1	P0DTD1
Nsp3	1945aa	ORF1a	YP_009742610.1	P0DTD1
Nsp13	601aa	ORF1a	NP_828870.1	P0DTD1
ORF7a	121aa	ORF7a	YP_009724395.1	P0DTC7
S	1273aa	S	YP_009724390.1	P0DTC2

**Table 2 microorganisms-11-02902-t002:** Self-proteins associated with IDDs and autoimmune encephalitis.

Protein	Number of Aminoacids	Gene	NCBI Reference Sequence	Uniprot ID
2’,3’-Cyclic-nucleotide 3’-phosphodiesterase (CNP)	421aa	CNP	NP_149124.3	P09543
Aquaporin-4 (AQP4)	323aa	AQP4	NP_001641.1	P55087
Glutamic acid decarboxylase 65-kilodalton isoform (GAD65)	585aa	GAD2	NP_001127838.1	Q05329
Myelin-associated glycoprotein (MAG)	626aa	MAG	NP_002352.1	P20916
Myelin basic protein (MBP)	304aa	MBP	NP_001020272.1	P02686
Myelin oligodendrocyte glycoprotein (MOG)	247aa	MOG	NP_996532.2	Q16653
Myelin-associated oligodendrocytic basic protein (MOBP)	183aa	MOBP	NP_001380633.1	Q13875
Myeloperoxidase (MPO)	745aa	MPO	NP_000241.1	P05164
N-methyl-D-aspartate receptor 1 (NMDAR1)	938aa	GRIN1	NP_015566.1	Q05586
Transaldolase	337aa	TALDO1	NP_006746.1	P37837

**Table 3 microorganisms-11-02902-t003:** Linear and three-dimensional identity between SARS-CoV-2 antigens and self-antigens, and antigenic properties of each combination.

SARS-CoV-2 Antigens	Autoantigens	Region of the SARS-CoV-2 Antigen with More Identity	Region of the Autoantigen with More Identity	% Identity	E-Value	SWISS MODEL SARS-CoV-2 Antigen	SWISS MODEL Autoantigen	TM-SCORE	RMSD	Overall Prediction Vaxijen Linear	Overall Prediction VaxiJen Three-Dimensional Model
M	NMDAR1	54–70	562–577	100	3.00 × 10^−4^	74–106	626–658	0.89	0.55	0.9306	0.5324
M	MPO	134–162	57–85	100	1.0	76–105	67–96	0.73	1.21	0.4740	0.5177
Nsp2	NMDAR1	448–465	325–344	71.43	0.038	549–584	621–650	0.69	1.35	0.5983	0.4174
S	MOG	249–278	83–102	83.33	0.069	944–974	150–180	0.63	2.06	0.4706	0.4059
ORF7a	MOG	25–32	95–102	75	2.00 × 10^−4^	17–81	30–153	0.62	2.83	0.4846	0.6598
N	MPO	227–236	149–157	70	0.002	388–419	67–98	0.59	1.91	0.4117	0.4124
Nsp13	GAD65	466–472	439–445	66.67	3.00 × 10^−4^	290–349	312–389	0.52	3.22	0.6555	0.4695
Nsp1	GAD65	131–138	137–139	100	0.005	32–61	302–331	0.52	1.57	−0.3299	0.6325
Nsp1	MOG	103–114	210–221	71.43	0.003	33–62	204–232	0.50	1.59	0.6013	0.6757
Nsp3	MPO	903–908	613–618	83.33	2.00 × 10^−5^	180–209	71–112	0.50	2.79	1.8236	0.8823
S	NMDAR1	1020–1027	223–230	100	0.017	1020–1050	221–250	0.50	1.47	0.8726	0.7359
Nsp3	NMDAR1	1800–1809	766–774	100	5.00 × 10^−4^	399–535	153–278	0.49	3.86	0.4238	0.4044
Nsp13	PLP	88–94	99–105	71.43	1.00 × 10^−4^	310–342	175–210	0.49	2.82	−0.0624	0.4685
Nsp1	PLP	34–62	196–211	100	0.006	33–64	240–272	0.47	1.69	0.7645	0.5854
Nsp1	Transaldolase	84–99	139–154	100	0.001	33–62	145–176	0.46	2.18	0.6898	0.6757
Nsp2	MPO	419–425	159–165	100	0.15	672–707	67–95	0.46	1.16	0.5813	0.5890
Nsp3	PLP	88–94	99–105	71.43	1.00 × 10^−4^	180–209	34–63	0.45	2.07	**−0.0624**	0.8823
M	MAG	132–138	369–375	100	2.00 × 10^−4^	156–186	295–325	0.43	2.59	0.4549	0.6527
S	Transaldolase	1110–1117	37–44	62.50	0.002	276–305	125–156	0.42	2.10	0.8734	0.6476
Nsp13	Transaldolase	146–151	307–312	83.33	0.001	367–396	135–164	0.41	2.60	1.0726	1.1104
N	CNP	243–252	191–200	100	0.001	195–239	322–363	0.41	2.72	0.7056	0.6601
Nsp13	NMDAR1	301–306	628–636	100	3.00 × 10^−4^	503–532	683–712	0.41	2.90	0.9060	0.4818
Nsp2	Transaldolase	389–403	95–109	66.67	0.19	249–280	134–163	0.40	1.90	0.5146	0.7524
Nsp2	CNP	270–278	348–356	77.78	4.00 × 10^−7^	395–432	349–382	0.39	2.38	0.9295	0.4387
N	MOBP	198–209	168–178	77.78	5.00 × 10^−6^	202–233	29–74	0.38	2.38	0.4291	0.6320
N	PLP	170–184	120–134	75	0.0003	212–241	75–98	0.37	1.44	0.4459	0.7169
Nsp1	MBP	76–102	198–116	100	0.010	32–67	211–240	0.36	3.22	0.6583	0.4770
Nsp3	AQP4	63–77	51–65	100	0.27	2667–2697	197–227	0.35	2.36	0.4168	0.5551
Nsp2	MBP	61–73	107–119	100	8.00 × 10^−4^	280–309	200–229	0.33	2.79	0.6625	0.5624

**Table 4 microorganisms-11-02902-t004:** The viral epitopes for MHC-I and MHC-II and the corresponding homologous human proteins.

SARS-CoV-2 Antigen	Autoantigens	Allele	Potential SARS-CoV-2 Epitope	Corresponding Human Epitope	IC50 Virus Peptide	IC50 Human Peptide
M	NMDAR1	HLA-DQA1*01:02/DQB1*06:02	NWITGGIAIAMACLV	VWAGFAMIIVASYTA	57.00	60.00
		HLA-DRB1*15:01	LMWLSYFIASFRLFA	GFAMIIVASYTANLA	68.00	49.00
		HLA-A*31:01	LSYFIASFR	LGMVWAGFAM	12.20	262.06
M	MPO	HLA-A*31:01	LSYFIASFR	KQLVDKAYK	12.20	68.78
		HLA-B*07:02	GGIAIAMACLV	RLRSGSASPM	321.23	159.89
Nsp2	NMDAR1	HLA-DQA1*01:02/DQB1*06:02	RVLQKAAITILDGIS	VWAGFAMIIVASYTA	93.00	60.00
		HLA-DRB1*15:01	ITILDGISQYSLRLI	VWAGFAMIIVASYTA	146.00	60.00
		HLA-A*31:01	RTLETAQNSVR	GAPRSFSAR	309.40	129.86
		HLA-B*07:02	SVRVLQKAAI	APRSFSARIL	357.95	29.30
S	MOG	HLA-DRB1*15:01	LNTLVKQLSSNFGAI	GVLVLLAVLPVLLLQ	89.00	58.00
		HLA-DQA1*01:02/DQB1*06:02	QNAQALNTLVKQLSS	WVSPGVLVLLAVLPV	319.00	229.00
ORF7a	MOG	HLA-DQA1*01:02/DQB1*06:02	YQECVRGTTVLLKEP	YWVSPGVLVLLAVLP	261.00	210.00
N	MPO	HLA-A*31:01	FSKQLQQSMSS	KQLVDKAYK	46.72	68.78
Nsp13	GAD65	HLA-DRB1*15:01	AIGLALYYPSARIVY	AKQKGFVPFLVSATA	71.00	196.00
		HLA-DQA1*01:02/DQB1*06:02	IVYTACSHAAVDALC	LVSATAGTTVYGAFD	347.00	52.00
		HLA-A*31:01	KYLPIDKCSR	KHKWKLSGVER	18.72	35.44
		HLA-B*07:02	LPIDKCSR	VPFLVSAT	187.50	69.82
Nsp1	GAD65	HLA-DRB1*03:01	SVEEVLSEARQHLKD	RGKMIPSDLERRILE	489.00	448.00
		HLA-A*31:01	HLKDGTCGLVE	KMIPSDLERR	31.44	27.31
Nsp1	MOG	HLA-A*31:01	HLKDGTCGLVE	CWKITLFVIVP	31.44	284.79
Nsp3	MPO	HLA-A*31:01	SYKDWSYSGQS	RLRSGSASPME	38.77	111.56
S	NMDAR1	HLA-DQA1*01:02/DQB1*06:02	ASANLAATKMSECVL	ASEDDAATVYRAAAM	213.00	121.00
		HLA-A*31:01	KMSECVLGQSKR	LSASEDDAATVYR	297.9	495.7
		HLA-DQA1*01:02/DQB1*06:02	AQYTSALLAGTITSG	SRRVLLLAGRLAAQS	122.00	199.00
		HLA-B*07:02	MIAQYTSAL	MAAESRRVL	61.88	54.29

## Data Availability

The results presented in this article are supported by data in other articles published in MDPI journals. The clinical profile and risk factors for severe COVID-19 in our cohort of hospitalized patients comparing the first and second pandemic waves were published in the *Journal of Clinical Medicine* in 2023, under the [88]. Disease severity was associated with older age, pre-existing neurological comorbidities, higher viral load, and dyspnea. Laboratory biomarkers related to white blood cells, coagulation, cellular injury, inflammation, and renal and liver injuries were significantly associated with severe COVID-19. During the second wave of the pandemic, the necessity of invasive respiratory support was higher, and more individuals with COVID-19 developed acute hepatitis, suggesting that the progression of the second wave resulted in an increase in severe cases. We used transcriptome analysis of these patients to understand key genes and cellular mechanisms that are most affected by the severe outcome of COVID-19. Transcriptomic analysis revealed 1009 up-regulated and 501 down-regulated genes in the SARS group, with 10% of both being composed of long non-coding RNA. Ribosome and cell cycle pathways were enriched among down-regulated genes. The most connected proteins among the differentially expressed genes involved transport dysregulation, proteasome degradation, interferon response, cytokinesis failure, and host translation inhibition. Furthermore, interactome analysis showed fibrillarin to be one of the key genes affected by SARS-CoV-2. This protein interacts directly with the N protein and long non-coding RNAs affecting transcription, translation, and ribosomal processes. This work was published in the *International Journal of Molecular Sciences* in 2022, under the [38]. We also published a study that aimed to establish a relationship between miRNA and neurological manifestations in our cohort of COVID-19 patients co-infected with HHV-6 and evaluate miRNAs as potential biomarkers. miRNA analysis by real-time polymerase chain reaction (qPCR) revealed miRNAs associated with neuroinflammation were highly expressed in patients with neurological disorders and HHV-6 detection. When compared with the group of patients without detection of HHV DNA and without neurological alterations, the group with detection of HHV-6 DNA and neurological alteration displayed significant differences in the expression of mir-21, mir-146a, miR-155, and miR-let-7b (*p* < 0.01). This work was published in the International Journal of Molecular Sciences in 2023, under the [89]. One of our authors (Salvio, AL) also analyzed the effectiveness of household disinfection techniques to remove SARS-CoV-2 from cloth masks. The study showed that all biocidal treatments successfully disinfected the tissue tested. This work was published in *Pathogens* in 2022, under the [90].

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
