# Peer review of "Molecular Mimicry between SARS-CoV-2 Proteins and Human Self-Antigens Related with Autoimmune Central Nervous System (CNS) Disorders"

_microorganisms, 2023, doi:10.3390/microorganisms11122902_

Round 1

Reviewer 1 Report

Comments and Suggestions for Authors

Line 25 antigens

Line 38 COVID-19

Line 45 Check references

Lines 107 and 245, “Three-dimensional”

Figure 1-3. Indicate with arrows the injuries described in the text

Figure 4 would be worth representing each protein larger, even showing an overlap of both proteins, the human protein (illustrated in one color) and the SARS-CoV-2 protein (another color). They can do this with the Chimera USCF software. Also, in this figure, I would add the sequence alignment for each protein.

Table 4 shows that the putative epitopes of SARS-CoV-2 are rich in Ala and Leu. Remarkably, these amino acids form loops, which is why, in Figure 4, all the putative epitopes are loops. Has anything been mentioned in the literature on other viruses or diseases and how this is associated with CNS damage?

Comments on the Quality of English Language

Minor editing of English language required

Reviewer 2 Report

Comments and Suggestions for Authors

The paper about possible molecular mimicry between SARS-CoV-2 antigens and CNS autoimmune self-antigens is actual and interesting for reading.  

The combining of clinical and morphological data with plenty of bioinformatics approaches is modern and give us new useful information.

There are minor points that should be addressed:

1. It would be nice to have a separate list of abbreviations used through the paper, especially the specific ones (e.g. various doseases and syndromes as well as bioinformatics algorithms, such as SMM-align).

2. Figures 1, 2, 3 should have more detailed descripion in the Figure legends.  It would be nice to see some arrows indicating the lesions and other morphological features/structures described in the text. 

3.  Line 94: França et al (2023) sould be indicated as a numbered reference.

4. Line 98: Please add into the phrase "based on results from PubMed..."  some main references which you rely on.   
